# Enhancement of Anaerobic Digestion of Corn Straw: Effect of Biological Pretreatment and Heating with Bio-Heat Recovery from Pretreatment

Shanyue Guan [1,2,3], Chao He [1,2,3], Pengfei Li [1,2,3,*], Panpan Li [1,2,3], Tingting Hou [1,2,3], Zan Gao [4], Gang Li [1,2,3] and Youzhou Jiao [1,2,3,5,*]

1 Key Laboratory of New Materials and Facilities for Rural Renewable Energy of Ministry of Agriculture and Rural Affairs, College of Mechanical & Electrical engineering, Henan Agricultural University, Zhengzhou 450002, China; guanshany@126.com (S.G.)
2 Henan International Joint Laboratory of Biomass Energy and Nanomaterials, Henan Agricultural University, Zhengzhou 450002, China
3 Henan Collaborative Innovation Center of Biomass Energy, Henan Agricultural University, Zhengzhou 450002, China
4 Zhongyuan Environment Zhihe (Zhengzhou) Water Environment Technology Company, Zhengzhou 450002, China
5 Henan University of Engineering, Zhengzhou 450002, China
* Correspondence: lpfbiomass@henau.edu.cn (P.L.); jiaoyouzhou@henau.edu.cn (Y.J.); Tel.: +86-18037381289 (P.L.); +86-13526637709 (Y.J.)

**Abstract:** Biological pretreatment can promote the degradation of biomass and enhance methane production via the subsequent anaerobic digestion. In addition, a large amount of bio-heat can be generated during the pretreatment process to provide heat for the anaerobic digestion process. In this study, composite microorganisms were employed for pretreating corn straw. The impact of different pretreatment times and the heat generated by the pretreatment process on subsequent anaerobic digestion were studied. The results show that the maximum temperature of the pretreatment process was 56.2 °C, obtained on day 6. After 14 days of pretreatment, the degradation rate of the pretreatment group increased by 41% compared with the control group. As a consequence, straws with different pretreatment times were used for anaerobic digestion. The group that underwent 6 days of pretreatment and utilized bio-heat generated from pretreatment achieved the highest cumulative methane production of 401.58 mL/g VS, which was 60.13% higher than in the control group without pretreatment. After 6 days of composite microorganism pretreatment, the group that utilized bio-heat achieved a 29.08% increase in cumulative methane production compared to the group that did not utilize bio-heat. In conclusion, this study highlights the potential of biological pretreatment with composite microorganisms followed by anaerobic digestion using bio-heat as an effective method for treating corn straw.

**Keywords:** composite microorganisms; pretreatment; bio-heat; anaerobic digestion

## 1. Introduction

Straw, the main waste in agricultural production, is a good raw material for anaerobic digestion. The straw can be effectively converted into biogas through anaerobic digestion, and this conversion supplements the energy supply for production and life, improves the utilization of straw resources, and promotes the development of traditional agriculture to modern ecological agriculture [1,2]. In recent years, suitable pretreatment methods, additives, and optimized processes have become the focus of study of enhanced anaerobic digestion in the field of biomass anaerobic digestion [3,4].

Straw biomass is not easily decomposed by the microorganisms associated with anaerobic fermentation because of its high lignocellulose content [5]. Due to the complex crystal structure of lignin, which has intertwined cellulose, hemicellulose, and lignin on its surface, the degradation of any one component is limited by the other components [6]. Some anaerobic bacteria have the potential to degrade lignin under natural conditions, but this degradation capacity is limited [7–9]. As a result, there are some problems in the anaerobic fermentation process, such as slow initiation of fermentation, low gas production efficiency, and long fermentation periods [10]. Pretreatment before anaerobic fermentation is therefore effective. The selection of corn straw pretreatment methods should be based on reducing costs, increasing gas production, shortening process time, and simplifying procedures [11]. The pretreatment methods that have been used by researchers in recent studies include physical, chemical, and biological methods. The energy consumption of the physical method is high, and the treatment effect is often closely related to the cost input. Although the chemical method is simple and convenient, it has certain requirements for equipment and has a problem with environmental pollution, which is not conducive to industrial application. Compared with the first two methods, the biological method has the advantages of low energy consumption and no environmental pollution, but it takes longer [12]. Enzymatic pretreatment and microbial pretreatment are widely used in biological methods. Compared with microbial pretreatment, the cost of enzymatic pretreatment is high, although the functionalization is simple. Due to the complexity of the lignocellulose structure, it is difficult to access the internal tissue structure of straw using enzymes only.

Composite microorganism pretreatment can be used for the biodegradation of straw biomass via the action of fungi and enzymes, taking advantage of its strong biodegradation ability [13]. It separates cellulose, hemicellulose, and lignin from each other so that anaerobic microorganisms are in full contact with the biomass, thus increasing the efficiency of gas production. It also plays a positive role in promoting anaerobic fermentation [14]. Zhu et al. [15] constructed the AMC (aerobic microbial consortium) aerobic complex strain, and the cumulative methane production from anaerobic fermentation was 233.09 mL·$g^{-1}$ VS and 242.56 mL·$g^{-1}$ VS after AMC and ANMC (anaerobic microbial consortium) biological pretreatment, respectively, which were increases of 6.89% and 11.23% compared with the control group. Sameh et al. [16] used new composite microorganisms to pretreat wood chips before anaerobic fermentation, increasing the cumulative gas production and methane production by 86.4% and 92.2%, respectively, compared with the control group. Zanellati et al. [17] extracted and screened composite microorganisms from unsterilized corn silage, and the cumulative methane production from the digestive liquid increased by 70% after pretreatment. A novel microbiome used by Ali et al. [18], CS-5 and BC-4, increased cumulative and methane production by 76.3% and 64.3%, respectively.

The composite microorganism pretreatment process, in which energy conversion occurs at all times, is a dynamic biochemical process realized by the joint action of multiple microbial communities [19]. However, current studies have focused more on the effect of lignocellulosic degradation on methane production and less on the accumulation of bio-heat energy during composite microorganism pretreatment. Some scholars have studied the thermal effect of microflora during growth and metabolism in other fields, but few studies have exploited this thermal effect to promote anaerobic fermentation. Xu et al. [20] studied the change in heat flow in the tank during the fermentation of corynebacterium glutamate using a calorimeter and qualitatively analyzed the relationship between the reaction process parameters and the change in heat flow. Zhuang et al. [21] studied the fermentation of dry red wines under a constant temperature environment and found that the temperature in the dry red wine fermenters increased significantly during the fermentation process. Klejment et al. [22] studied the heat production and thermal conductivity of municipal waste composting during the high-temperature stage and obtained an average heat release of 1136 KJ/kg. Bialobrzewski et al. [23] investigated the mathematical model of microbial heat production capacity at medium and high temperatures during the composting of

sewage sludge straw. In our previous works, we constructed a composite of microorganisms and found that the strain exhibited strong thermal effects during the pretreatment of crop straw, and the component structure of the straw changed correspondingly. However, the superposition of bio-heat utilization and lignocellulosic degradation to enhance anaerobic digestion was neglected in subsequent studies.

Based on the previous studies, bio-heat accumulation during the pretreatment of composite microorganisms was investigated and the promotion of anaerobic digestion by bio-heat was studied. The results provide a reference for effective connection and matching performance between biological pretreatment and anaerobic digestion.

## 2. Materials and Methods

### 2.1. Materials

The composite microorganisms used in this study were screened and cultured at the Key Laboratory of New Materials and Facilities for Rural Renewable Energy of the Ministry of Agriculture and Rural Affairs. Among them, *Gloeophyllum trabeum*, *Bacillus circulans*, *Streptomyces maritimus,* and *Pseudomonas aeruginosa* were purchased from China General Microbial Species Preservation and Management Center, while *Phanerochaete chrysosporium*, *Trichoderma viride*, *Aspergillus niger,* and *Coriolus versicolor* were provided by the College of Life Science, Henan Agricultural University. The composite microorganisms were preserved after antagonistic tests and expansion.

Corn straw was obtained from the experimental field of Henan Agricultural University. The harvested corn straw was dried under natural conditions and crushed to 40 mesh by the grinder. Its components are shown in Table 1.

**Table 1.** Physicochemical properties of corn straw and inoculum.

|  | C (%) | N (%) | C/N (%) | Cellulose (%) | Hemicellulose (%) | Lignin (%) | TS (%) | VS (%) |
|---|---|---|---|---|---|---|---|---|
| Corn straw | 49.91 | 1.58 | 31.59 | 43.06 | 19.24 | 17.55 | 92.66 | 81.20 |
| Inoculum | 24.06 | 4.10 | 5.87 | / | / | / | 8.33 | 2.58 |

The inoculum was obtained after anaerobic digestion experiments on municipal sludge at the Key Laboratory of New Materials and Facilities for Rural Renewable Energy of the Ministry of Agriculture and Rural Affairs. The characteristics of the inoculum are shown in Table 1.

### 2.2. Experimental Set-Up

In the pretreatment phase, the composite microorganisms were inoculated into a liquid culture medium. Next, they were placed in a constant temperature oscillating incubator for 2 days of continuous activation (30 °C and 120 r/min). Among these, *Bacillus circulans* was cultured in nutrient gravy medium, *Pseudomonas aeruginosa* was cultured in LB medium, *Streptomyces maritimus* was cultured in a starch medium, and *Phanerochaete chrysosporium*, *Coriolus versicolor*, *Trichoderma viride*, *Aspergillus niger,* and *Gloeophyllum trabeum* were cultured in PDA (potato dextrose agar) medium. Then, 50 g of corn straw was added according to a solid–liquid mass ratio of 5:1, and the ratio of each strain was 1:1:1:1:1:1:1:1:1:1. After this, the mixture of composite microorganisms and corn straw was added into a vacuum glass device after mixing well. Pretreatment of the composite microorganisms was carried out in the incubator for 14 days, with an initial temperature of 25 °C, a set initial pH of 8, water content of 15%, and 2.8 h of ventilation per day with a flow rate of 0.011 m$^3$/h. The humidity in the incubator was maintained above 80%. The group not inoculated with composite microorganisms was set as the control group.

To examine the impact of varying pretreatment durations and the resultant heat generation on anaerobic digestion, the anaerobic digestion experiment was segmented into 14 runs, each corresponding to a different number of pretreatment days prior to anaerobic digestion. The pretreatment experiments were conducted in vacuum bottles. Following

pretreatment, the inoculum was promptly introduced into the bottles, and deionized water was used to regulate the TS concentration to 4%. The bottles were then sealed for anaerobic digestion trials. Subsequently, the vacuum bottles were placed in a 35 °C incubator for a 30-day anaerobic digestion process. This ensured that the material and bio-heat reached the anaerobic digestion stage. These 14 runs were set up as the pretreatment group. A control group was then set up with the same 14 groups and carried into the anaerobic digestion stage. After cooling at ambient temperature (18 ± 2 °C), these 14 runs were set up as the cooling group. The cooling group with 2, 4, 6, 8, and 10 d of pretreatments was selected and used to compare the effect of gas production with pretreatment groups collected on the same pretreatment days.

### 2.3. Modified Gompertz Equation

The modified Gompertz model was used to analyze the kinetics of anaerobic digestion of corn straw after composite microorganism pretreatment.

The dynamic analysis model formula is represented by Equation (1):

$$B = P \exp\left(-\exp\left(\frac{R_m e}{P}(\lambda - t) + 1\right)\right) \tag{1}$$

$B$ is the cumulative methane production rate (mL/g VS) at digestion time $t$ (d), $P$ is the maximum methane production (mL/g VS), $R_m$ is the maximum methane production rate (mL/(g d) VS), $\lambda$ is the lag time (d), and $e$ is the mathematical constant 2.718. The parameters of the Gompertz model were obtained using nonlinear curve fitting in the software Originpro 8.6.

### 2.4. Analytical Methods

A 2 mL volume of pretreatment solution was taken from the pretreatment unit every 2 days. Next, the pretreatment solution was centrifuged at 5000 rpm for 10 min, then the supernatant was measured with a COD meter (KHCN-200A, Nanjing Kehuan, Nanjing, China). Scanning electron microscopy (SEM) using a HITACHI S-3400 N II (Hitachi, Ltd, Tokyo, Japan) was used to examine the changes in the physical structure of corn straw.

According to the standard method [24], the digestion liquid was dried at 105 °C to a constant weight to calculate the sample TS. The sample was then put into a muff furnace and burned at 600 °C for 3 h. After this, the sample's volatile solids (VSs) were weighed. VFAs and methane contents in the biogas were determined using gas chromatography (Agilent 6820C, Agilent Technologies Inc., Santa Clara, CA, USA). $N_2$ was used as the carrier gas, the TCD detector was set to 150 °C, and a 5A molecular sieve was used to fill the column at a retention time of 2.0 min.

The cellulose, hemicellulose, and lignin contents of corn straw were determined using the paradigm water washing method as employed by Zhang [25]. Their initial contents in corn straw were measured before the experiment, and then daily during the pretreatment process. The degradation rate was defined by dividing the amount of degraded substance by the total amount of substance (in percentage). The methane content was analyzed using an Agilent GC-6820 (Agilent Technologies Inc., Santa Clara, CA, USA) gas chromatograph. The analytical column was a Porapak Q gas-packed column. The column temperature was set to 60 °C, while the temperature of the thermal conductivity detector was set to 80 °C. The flow rate of the carrier gas $N_2$ was set to 25 mL/min.

### 3. Results

#### 3.1. Temperature Variations

It can be seen from Figure 1 that the temperatures of both the composite microorganisms group and the control group increased from the start of the pretreatment and were much higher than the ambient temperature. The highest temperature of the composite microorganisms pretreatment group reached 56.2 °C on day 5. Compared to the control group, the highest temperature in the composite microorganisms group increased by 27.14%. The

warming period saw a rapid growth and metabolism of the composite microorganisms, leading to their adaptation to the environment. The bacterial community generated a large amount of metabolic heat while degrading the biomass, and the accumulation of heat in turn increased the metabolic rate of microorganisms during pretreatment. Throughout the 14 days of pretreatment, the temperature of the composite microorganisms group was always higher than that of the control group, indicating that the metabolic activities of the microorganisms in the composite microorganisms group were more rapid. After the warming period, the temperature began to decrease. The reason for this was that the increasingly high temperature inhibited the growth and metabolism of mesophilic bacteria, while on the other hand, the consumption of degradable substrates and the counter-inhibitory effect of metabolites reduced microbial activity. However, the control group experienced a relatively stable high-temperature period (above 40 °C) after the warming period, with a slight drop in temperature after day 5, followed by a small boost on days 8–10. This also indicated that the degradation of corn straw without the intervention of composite microorganisms is slow and steady.

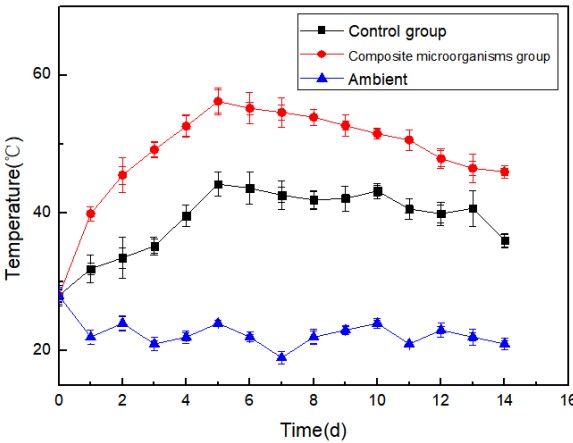

**Figure 1.** Temperature variations during the pretreatment of composite microorganisms.

*3.2. Variations in the Components of Corn Straw*

Changes in the contents of cellulose, hemicellulose, and lignin during the pretreatment process are shown in Figure 2. It can be seen from the solid line that the degradation rate of corn straw changed most significantly in the first 3 days: the degradation rates of hemicellulose and cellulose reached 55% and 40%, respectively, while the degradation rate of lignin was only 2%. Cellulose was surrounded by lignin and hemicellulose, but the degradation rate of cellulose was higher than that of lignin. The reason for this phenomenon is that the pretreatment had just started and the temperature did not reach the temperature at which the thermophilic lignin-degrading bacteria were adapted; as a result, the activity of lignin-degrading bacteria was not strong, and their metabolic capacities were weak. This indicated that bio-heat had a great influence on the metabolic capacities of bacteria and the sequence of degradation of cellulose, hemicellulose, and lignin. It is obvious from the dotted line that the sequence of degradation was also verified in the control group. In the control group, the degradation rate of lignin was higher than those of cellulose and hemicellulose in the first 6 days. The degradation rate of the composite microorganisms pretreatment group was 37.5% higher than that of the blank control group.

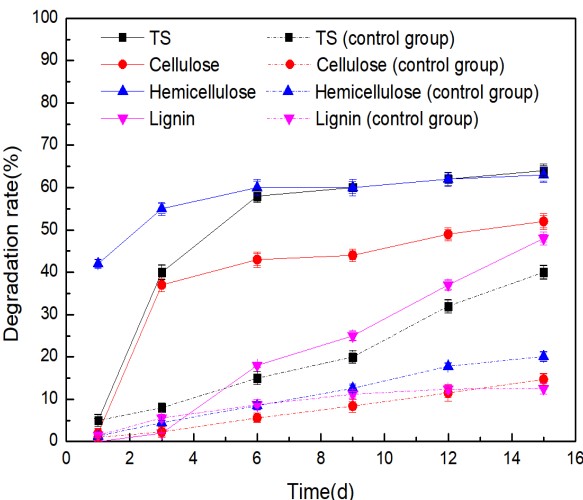

**Figure 2.** Variations in the components of corn straw during the pretreatment of composite microorganisms.

### 3.3. Variations in COD and VFAs

As can be seen in Figure 3a, the COD values showed an increasing trend from day 0 to day 6 during the pretreatment period, and the maximum value of 9600 mg/L was reached on day 6. After the peak, the COD value decreased and finally stabilized. The COD values of the control group increased slowly and fluctuated weakly, but were always lower than those of the composite microorganisms group during the pretreatment period. This pattern is consistent with the conclusions of Yuan [26].

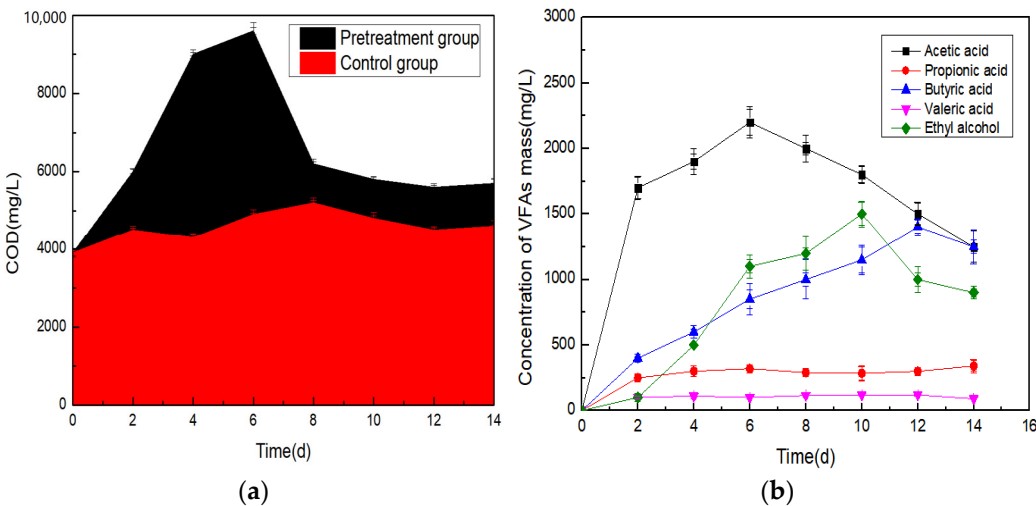

| (a) | (b) |

**Figure 3.** Variations in COD and VFAs during composite microorganisms pretreatment: (**a**) Variations in COD during composite microorganisms pretreatment; (**b**) Variations in VFAs during composite microorganisms pretreatment.

The changes in VFAs are shown in Figure 3b. Acetic acid accumulated rapidly in the first 6 days, reaching 2200 mg/L, and then began to decrease. Propionic acid and valeric acid contents remained at low levels and almost stable throughout the whole pretreatment period. Butyric acid and ethanol showed an increasing trend; ethanol reached a maximum value on day 10 and butyric acid reached a maximum value on day 12. It can be determined from the trends of COD and VFAs that on the 6th day of pretreatment, the content of soluble organic matter was higher. These could be used by methanogens to a higher extent, and the substrate had the highest potential for anaerobic digestion. Previous studies have shown that VFAs are important in the $CH_4$ metabolic chain and accumulate rapidly at

the beginning of biological pretreatment [27]. Therefore, the VFA values can be used as references for the methane production potential of corn straw during the composite microorganisms pretreatment period.

### 3.4. SEM of Corn Straw

The SEM of corn straw during the pretreatment process is shown in Figure 4. The surface of corn straw was flat and the structure was dense before pretreatment with composite microorganisms. This is consistent with Islam's observation [28]. As the pretreatment proceeded, the surface structure of corn straw was lumpy and part of the structure was deciduous. On the 2nd day of pretreatment, some cracks and holes appeared on the surface of the corn straw. Then, the cracks became larger and the holes bulged. As the surface of corn straw became rough and fluffy, the structure of lignocellulose changed, thereby increasing the exposed area of the cellulose [29]. By the 6th day of pretreatment, the internal structure of the corn straw was largely exposed. The results showed that the degradation enzymes produced by composite microorganism communities could destroy the surface coating of corn straw quickly and effectively. Under this condition, the composite mycelia extended into the corn straw through the small pores on the surface, and the internal lignocellulose was also degraded by the enzyme.

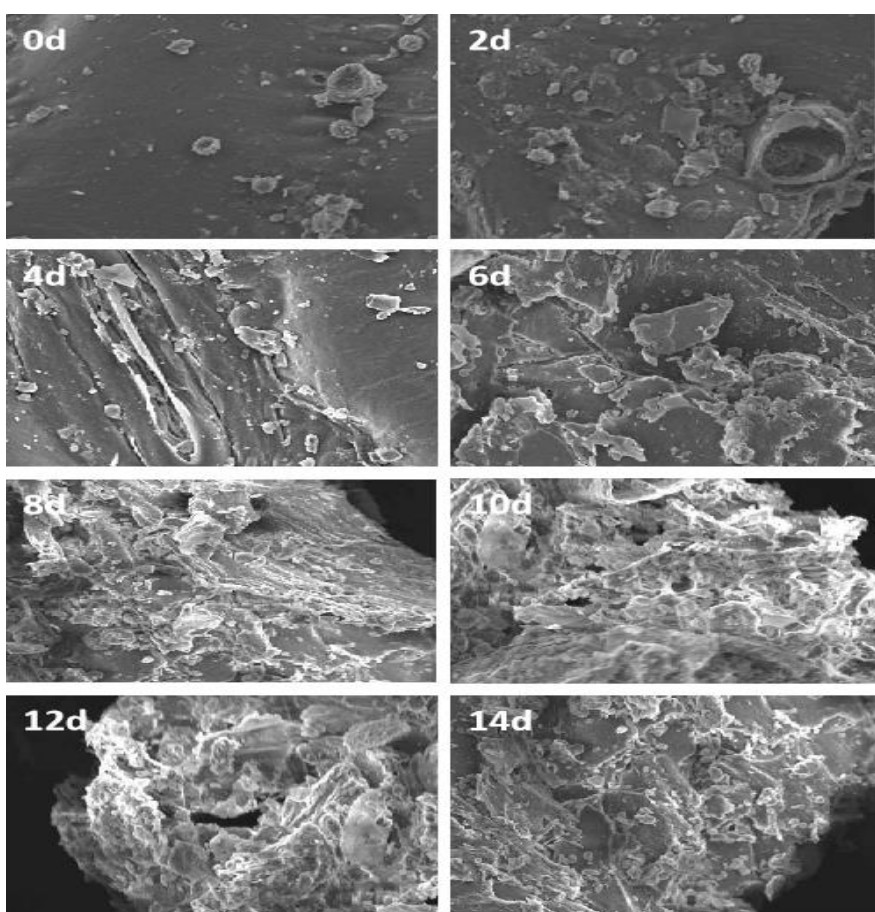

**Figure 4.** SEM of corn straw during the pretreatment of composite microorganisms (2 μm).

### 3.5. Gas Production via Anaerobic Digestion

As shown in Figure 5, methane production from corn straw after composite microorganism pretreatment was characterized. It can be seen in Figure 5a that in the 8 d pretreatment group, due to the longer pretreatment time and the higher accumulation of soluble organic matter, the peak of gas production appeared first on the 6th day of fermentation, with the highest daily gas production being 33.30 mL/g VS. This also validates the results

in Figure 3, where the VFAs and COD reached their maximum values on the 6th day. At this point, gas production levels in the day 3 and day 9 groups were similar to that in the day 8 group. However, gas production in the day 4 group was lower than in the other groups because of its lower pretreatment temperature compared to the day 9 group, resulting in less bio-heat accumulation, less active metabolism of the composite microorganisms, and less decomposition of soluble organic matter. Compared to the day 2 and day 3 groups, the reason for lower gas production in the day 4 group is that the acetic acid in the pretreatment substrate of the day 2 and day 3 groups was much higher than other molecular acids and was the main metabolic substrate of methanogens. As the pretreatment progressed, other molecular acids gradually accumulated and formed abundant metabolic substrates, which inhibited the metabolic pathway of methanogens to some extent. This was the reason why the peak of gas production in the day 6 group was one day later than that in the day 2 group. The maximum daily gas production in the day 6 group reached 46.305 mL/g VS on the 9th day, which was also the maximum gas production for the entire digestion period. The peak of gas production appeared 4 days earlier than in the blank control, and the maximum daily gas production increased by 55.26%. Some studies also showed that gas production via anaerobic digestion was significantly increased and the digestion period was shortened after pretreatment with composite microorganisms [30,31]. When the pretreatment time exceeded 10 days, gas production via anaerobic digestion efficiency decreased significantly. This was related to the excessive consumption of organic matter by the composite microorganisms in the pretreatment period.

As shown in Figure 5b, the methane content in each group increased continuously during the first 7 days of anaerobic digestion. The methane content in the day 2, day 4, and day 8 groups slowly decreased with digestion time, and by day 15 the methane content was already lower than in the blank control group. The methane content of the day 6 group decreased after the 7th day but remained at a high level, with the highest methane content reaching 70% and the average methane content remaining at 63.67% during the peak period. It can also be seen from Figure 5d that the corn straw pretreated with the composite microorganisms for 6 days had a higher methanogenic capacity. In Figure 5c, the overall gas production effect of the cooling group was lower than that of the non-cooling group. Among all cooling groups, the peak gas production was highest in the day 6 pretreatment group, reaching 33 mL/g VS; this was 28.73% lower and 3 days later than the non-cooling group. After a 30-day anaerobic digestion period, the cumulative methane production in the cooling group was significantly lower than in the non-cooling group at each pretreatment time. The difference in maximum cumulative methane production between the non-cooling and cooling groups was 29.08% over the 6-day pretreatment cycle. This indicated that the bio-heat from pretreatment has a significant contribution to the gas production of anaerobic digestion. During the initial stages of anaerobic digestion, this bio-heat plays an important role in the initiation of anaerobic digestion by methanogens. A comparison of the two blank control groups showed that the cooling group produced about the same amount of methane as the non-cooling group, which was due to the fact that residual heat in the non-cooling group at the pretreatment stage without the addition of the composite microorganisms were approximately the same as the start-up temperature of anaerobic digestion and therefore had little effect on the enhancement of gas production during anaerobic digestion. Compared to the control group, cumulative methane production in the cooling group after 6 days of pretreatment was still 43.78% higher even without heat. This shows the superimposed effect of biomass degradation and bio-heat on the enhancement of gas production from anaerobic digestion.

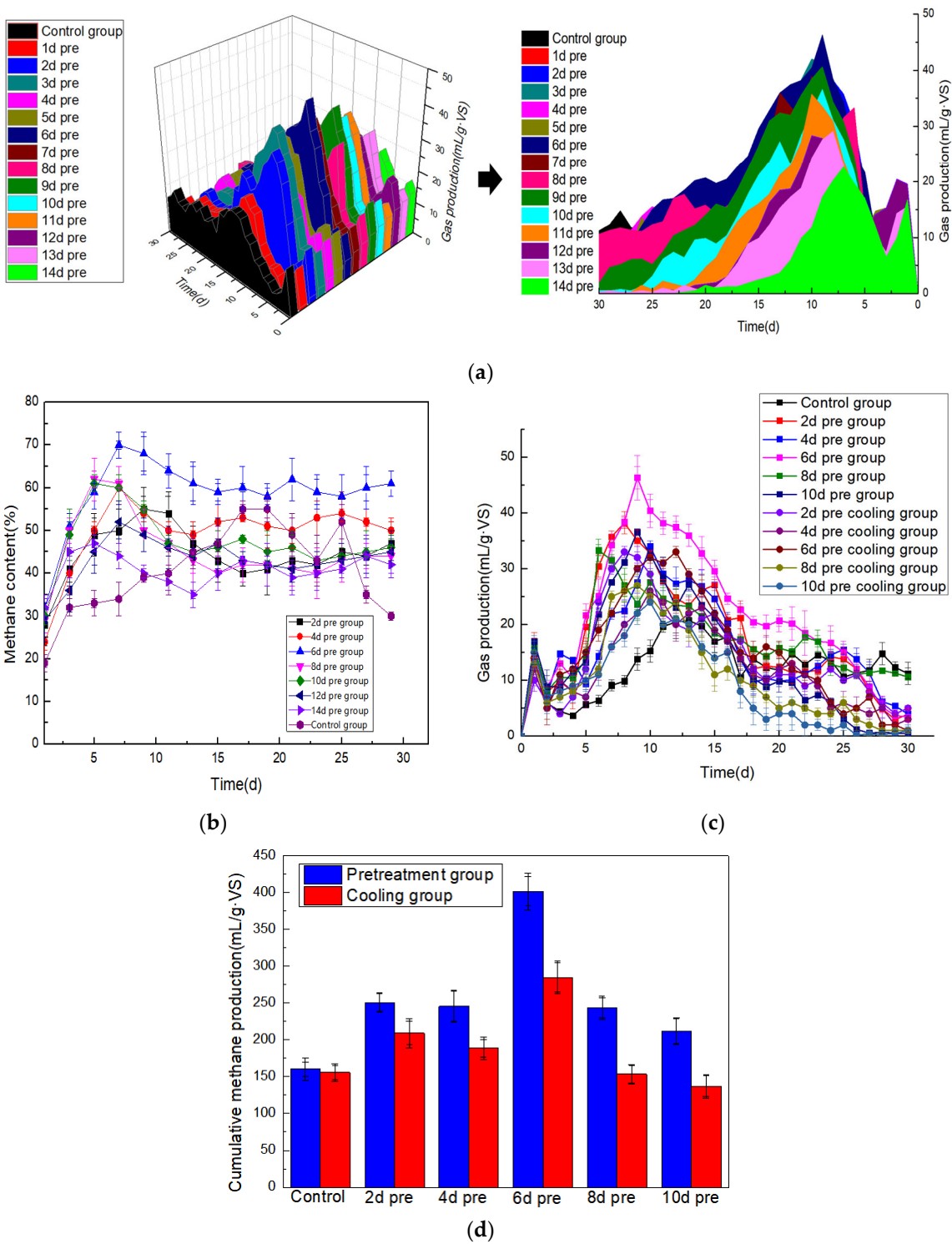

**Figure 5.** Gas production pattern of corn straw anaerobic digestion after composite microorganisms pretreatment: (**a**) Plots of daily gas production; (**b**) Plots of methane content; (**c**) Plots of daily gas production in the cooling group; (**d**) Plots of cumulative methane production.

*3.6. Dynamic Analysis*

The Gompertz model can be used to assess the potential of anaerobic digestion for methane production and the lag period of gas production. The modified Gompertz model was adopted to carry out kinetic analysis of the anaerobic fermentation of corn straw after composite microorganisms pretreatment. *B*, $R_m$, and $\lambda$ were obtained, as shown in Table 2.

From the model fitting parameters, it could be seen that the correlation coefficient $R^2$ of the modified Gompertz model of all groups was higher than 0.99. This indicates that the fitting was good and fitting parameters are reliable. The maximum cumulative gas production P of the 6 d pre group was 686.66 mL/g VS, which was the highest among all experimental groups, and the absolute deviation of this result from the actual value is only 5.0%. For the first 9 days of the pretreatment period, the maximum cumulative gas production P showed three peaks at 3, 6, and 9 days. The other groups were lower than that of the three groups. After the 9th day of pretreatment, the maximum cumulative gas production P of the groups began to decrease; the lowest value was 187.52 mL/g VS. The maximum gas production rate *Rm* was also consistent with this pattern. Among all the experimental groups, the maximum Rm value reached 38.21 mL/(g d) VS for the 9 d pre groups. The methane production delay time for all the experimental groups is 1.64–5.03 d, and the delay time for the blank control is 4.56 d. Only the 1 d pre-group was longer than the blank control group.

**Table 2.** Fitting parameters of the modified Gompertz model.

| Group Number | P (mL/g VS) | $R_m$ (mL/(g d) VS) | λ(d) | $R^2$ | Absolute Percent Deviation (%) |
|---|---|---|---|---|---|
| Control group | 459.63 | 16.91 | 4.56 | 0.9972 | 21.18 |
| 1 d pre group | 408.78 | 17.63 | 5.03 | 0.9976 | 15.06 |
| 2 d pre group | 550.53 | 31.27 | 2.95 | 0.9977 | 2.76 |
| 3 d pre group | 597.35 | 35.62 | 3.36 | 0.9977 | 1.38 |
| 4 d pre group | 506.19 | 26.19 | 3.13 | 0.9966 | 4.55 |
| 5 d pre group | 487.05 | 26.10 | 3.55 | 0.9953 | 5.12 |
| 6 d pre group | 686.66 | 38.21 | 3.64 | 0.9989 | 5.00 |
| 7 d pre group | 505.16 | 32.64 | 3.46 | 0.9978 | 1.28 |
| 8 d pre group | 560.04 | 25.06 | 2.41 | 0.9965 | 6.87 |
| 9 d pre group | 561.42 | 35.63 | 3.49 | 0.9993 | 2.64 |
| 10 d pre group | 441.48 | 30.94 | 3.27 | 0.9989 | 2.91 |
| 11 d pre group | 375.34 | 30.33 | 3.25 | 0.9967 | 2.40 |
| 12 d pre group | 317.50 | 25.03 | 1.80 | 0.9911 | 3.20 |
| 13 d pre group | 281.95 | 25.08 | 2.60 | 0.9944 | 1.75 |
| 14 d pre group | 187.52 | 20.56 | 1.64 | 0.9957 | 0.77 |
| 2 d cooling group | 453.39 | 25.98 | 3.37 | 0.9966 | 1.88 |
| 4 d cooling group | 391.87 | 22.13 | 3.76 | 0.9991 | 5.34 |
| 6 d cooling group | 485.10 | 29.98 | 3.67 | 0.9994 | 4.09 |
| 8 d cooling group | 329.50 | 23.74 | 3.32 | 0.9984 | 1.07 |
| 10 d cooling group | 287.80 | 21.60 | 3.17 | 0.9948 | 3.00 |

A comparison of the kinetic parameters of the cooling group and pretreatment group showed that the maximum cumulative gas production values of the cooling groups were all lower than those of the pretreatment groups. The methane production delay times in the cooling groups were more even, ranging from 3.17 to 3.76 d; this was also shorter than the blank control group. Both the pre-groups and the cooling group had the highest potential for methane production on day 6 of pretreatment, and the delay in methane production was shortened by 0.92 d and 0.89 d, respectively.

## 4. Discussion

The results of previous studies have shown that corn straw still contains a large number of lignocellulose residues, especially lignin, after the whole anaerobic digestion process, which is difficult for methanogens to degrade. Prior to anaerobic digestion, lignocellulose is degraded by composite microorganisms into small molecules that can be digested by methanogens, and this pretreatment method can improve methane production from corn straw. A large amount of bio-heat is generated during the process of composite microorganisms pretreatment, and this bio-heat can be used in the anaerobic digestion of corn straw,

thus contributing to the initiation of anaerobic digestion and reducing energy consumption. However, from the perspective of the first law of conservation of energy, the excessive consumption of lignocellulose and the generation of bio-heat during the pretreatment process can inhibit the subsequent anaerobic digestion. How to balance the relationship between bio-heat and methane production by regulating the pretreatment of composite microorganisms was the focus of this study. In previous experimental studies, we identified composite microorganisms with a strong capacity for lignocellulose degradation. In this paper, corn straw pretreatment tests were carried out using composite microorganisms, and the subsequent anaerobic digestion was explored. The synergistic effect of degradation rate and bio-heat on subsequent anaerobic digestion under different pretreatment times was investigated. The comparative experiments in the non-cooling and cooling groups were also discussed to verify the synergistic effect of bio-heat. From the results of the test, it can be seen that the temperature of the composite microorganisms group increased significantly during the pretreatment period. The degradation rate of lignocellulose increased with increasing temperature, and this increase showed a certain order that was related to the thermophilic range of different strains of the composite microorganisms. Corn straw had a maximum methanogenic potential when composite microorganism pretreatment proceeded to day 6. At this time, the bio-heat and corn straw substrates brought into the anaerobic digestion stage resulted in maximum methane production. This conclusion offers a reference for the industrial application of the composite microorganisms pretreatment method.

## 5. Conclusions

In this paper, the effects of different composite microorganisms' pretreatment time on the methanogenic performance of anaerobic digestion were investigated; the effect of the promotion of bio-heat on anaerobic digestion was also studied. The results showed that on the 6th day of pretreatment, the temperature reached a maximum value of 56.2 °C, which is an increase of 27.14%, and the degradation rate of corn straw increased by 41% compared to the control group in the same period. The day 6 pretreatment group had the highest potential for methane production, with a maximum daily gas production of 46.305 mL/g VS, an increase of 55.26% compared to the blank control. The peak gas production was 4 days earlier, the duration of peak gas production was 5 days longer, and the cumulative methane production increased by 60.13% compared with the control group. Comparison with the cooling group showed that anaerobic digestion of methane production was significantly better in the non-cooling group following the introduction of bio-heat. The peak of daily gas production was 36.3% higher than in the cooling group, and the cumulative methane production was 29.08% higher. On the other hand, the cumulative methane production in the cooling group was still 43.78% higher than in the control group that was not pretreated with the composite microorganisms.

**Author Contributions:** Conceptualization, Y.J. and C.H.; methodology, S.G., P.L. (Pengfei Li) and P.L. (Panpan Li); investigation, T.H. and G.L.; resources, Z.G.; writing-original draft preparation, S.G.; writing-review and editing, P.L. All authors have read and agreed to the published version of the manuscript.

**Funding:** This research was supported by the National Key Research and Development Program of China (No. 2023YFE0106000), the National Natural Science Foundation of China (No. 52176184), and the Science and Technology Project of Henan Province (No. 212102310328; No. 212102110222).

**Institutional Review Board Statement:** Not applicable.

**Informed Consent Statement:** Not applicable.

**Data Availability Statement:** The original contribution presented in the study are included in the article, further inquiries can be directed to the corresponding authors.

**Conflicts of Interest:** The author Zan Gao is employed by Zhongyuan Environment Zhihe (Zhengzhou) Water Environment Technology Company. The author declare that he has no competing financial interests or personal relationships that may have influenced the work reported in this study.

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
