# Peer review of "Enhancement of Anaerobic Digestion of Corn Straw: Effect of Biological Pretreatment and Heating with Bio-Heat Recovery from Pretreatment"

_fermentation, doi:10.3390/fermentation10030160_

Round 1

Reviewer 1 Report (New Reviewer)

Comments and Suggestions for Authors

The article presents an interesting experimental study on the effect of the pretreatment with microorganisms on the anaerobic digestion of corn straw. In some parts of the paper, the English language requires revision to make the sentences more clear. In the following lines are reported some aspects to be improved or corrected:

Page 2, lines 71-72: please explain the acronyms AMN and ANMC

Page 2, line 79: typewrite error “uesd”

Page 3, line 134: Please explain the acronym PDA

Page 3-4, lines 142-152: the described operations are not clear, could you explain better?

Page 4, line 160: “demurrage time” should be “lag time”

Page 4, lines 162-164: revise English language

Page 5, paragraph 3.2: It should be useful to define the degradation rate to make more clear the meaning of figure 2

Page 5, figure 2: there is a typewrite error in the y-axis label

Page 6, line 230: typewrite error “uesd”

Page 9, line 331: typewrite error “prodcution”

In the discussion about the 6d pretreated group it could be also mentioned the results reported in figure 3 that show that at day 6 there is a maximum of produced VFAs and COD that could also justify the maximum in production of methane in successive anaerobic digestion.

After these revisions, I think that the paper could be published.

Comments on the Quality of English Language

The paper is sometimes well written and sometimes require revision of the English language. A revision of the language is necessary

Author Response

Dear reviewer of Fermentation,

  Thank you for your valuable comments on manuscript. We have revised the manuscript accordingly based on your review comments. Corresponding revises have been highlighted in red in the manuscript. The following amendments have been made to your comments:

  • Page 2, lines 71-72: please explain the acronyms AMCand ANMC

Revision: The acronyms AMC was interpreted by the authors of the reference to be aerobic microbial consortium, and the acronyms ANMC was interpreted to be anaerobic microbial consortium. Specific references can be found in reference 15.

Line 72-75, “Zhu et al. [15], constructed the AMC(aerobic microbial consortium) aerobic complex strain, and the cumulative methane production of anaerobic fermentation was 233.09mL·g-1 VS and 242.56mL·g-1 VS after AMC and ANMC(anaerobic microbial consortium) biological pretreatment respectively, which increased by 6.89% and 11.23% compared with the control group.”

  • Page 2, line 79: typewrite error “uesd”

Revision:The typewrite has been corrected.

Line 82, “A novel microbiome used by Ali et al. [18],”

  • Page 3, line 134: Please explain the acronym PDA

Revision: The acronyms PDA was interpreted to be potato dextrose agar.

Line 136, “Phanerochaete chrysosporium, Coriolus versicolor, Trichoderma viride, Aspergillus niger and Gloeophyllum trabeum were cultured in PDA(Potato dextrose agar) medium.”

  • Page 3-4, lines 142-152: the described operations are not clear, could you explain better?

Revision: The manuscript has been redescribed in the experimental operations and corrections have been made at the appropriate places in the article.

Line 145-159, the sentence has been revised to “To examine the impact of varying pretreatment durations and the resultant heat generation on anaerobic digestion, the anaerobic digestion experiment was segmented into 14 groups, each corresponding to a different number of pretreatment days prior to anaerobic digestion. The pretreatment experiments were conducted using vacuum bottles. Following the pretreatment, inoculum was promptly introduced into the bottles, and deionized water was employed to regulate the TS concentration to 4%. Subsequently, the bottles were sealed for anaerobic digestion trials. Subsequently, the vacuum bottles were placed in a 35℃ incubator for a 30-day anaerobic digestion process. In this way it can be ensured that the material and bio-heat was brought to anaerobic digestion period. This 14 groups were set up as the pretreatment group. Then set up a control group with the same 14 groups which was carried into the anaerobic digestion stage after cooling in ambient temperature(18±2℃), this 14 groups were set up as the cooling group.The cooling group with 2,4,6,8,10 d of pretreatment was selected to compare the gas production effect with the pretreatment groups in the same pretreatment days. ”

  • Page 4, line 160: “demurrage time” should be “lag time”

Revision: The misdescription has been corrected in the manuscript.

  • Page 4, lines 162-164: revise English language

Revision: The English language of the corresponding position has been revised.

Line 165-169, the sentence has been revised to “B is the cumulative methane production rate (mL/g VS) at the digestion time t (d), P is the maximum methane production (mL/g VS), Rm is the maximum methane production rate (mL/(g d) VS),  is the lag time (d), and e is the mathematical constant of 2.718. The parameters of the Gompertz model were obtained by the software Originpro 8.6 by nonlinear curve fitting. ”

  • Page 5, paragraph 3.2: It should be useful to define the degradation rate to make more clear the meaning of figure 2

Revision: The definition of degradation rate has been added in section 2.4 of the manuscript.

Line 184-187, the added sentence “Initial content of them in corn straw was measured before experiment, and the content was measured daily during pretreatment process. The degradation rate was defined by dividing the amount of degraded substance by the total amount of substance(in percent).”

  • Page 5, figure 2: there is a typewrite error in the y-axis label

Revision: The typewrite error in figure 2 has been corrected.

Line 233, the figure 2 has been revised to

  • Page 6, line 230: typewrite error “uesd”

Revision: The typewrite has been corrected.

Line 248, “the content of soluble organic matter was higher, which could be used by methanogens”

  • Page 9, line 331: typewrite error “prodcution”

Revision: The typewrite has been corrected.

Line 405, “The methane production delay time of cooling groups were more even, ranging from 3.17-3.76 d”

  • In the discussion about the 6d pretreated group it could be also mentioned the results reported in figure 3 that show that at day 6 there is a maximum of produced VFAs and COD that could also justify the maximum in production of methane in successive anaerobic digestion.

Revision: Additional discussion has been provided at the appropriate places in the manuscript.

Line 278-281, the sentence has been revised to “the peak of gas production appeared firstly on the 6th day of fermentation, with the highest daily gas production of 33.30 mL/g VS, this also validates the results in Figure 3 where the VFAs and COD generated on 6th day reached their maximum values.”

Reviewer 2 Report (New Reviewer)

Comments and Suggestions for Authors

The manuscript assesses the influence of biological pretreatment and heating on the subsequent anaerobic digestion of corn straw. The pretreatment is performed by composite microorganisms in order to separate cellulose, hemicellulose and lignin apart. The heat generated by the pretreatment is used for anaerobic digestion. The authors focus on finding the duration of the pretreatment that leads to the highest biogas production. The evaluation is done with respect to a control group (no pretreatment) and cooling groups (with pretreatment but cooled to 18 degrees before used for anaerobic digestion). The topic is interesting and of importance as lignocellulosic biomass needs pretreatment before it is used for anaerobic digestion. Moreover, biological pretreatment is not commonly used. In spite of a methodology well developed, the interpretation of the results is sometimes not precise.

 -        The paper is about the treatment of corn straw, hence the conclusion in the abstract (lines 32-34) cannot be about “stabilizing corn straw” but about its treatment.

-        Lines 39-40: The sentence is not correct as the straw is converted to biogas through anaerobic digestion.

-        Lines 71-73: please define AMC and ANMC

-        Lines 98-99: The name and affiliation of one of the authors of the present study is not the proper way of referencing. Unless there is a publication that can be cited, the sentence should be reformulated in a way similar to “In their previous work, the authors/ we …..”

-        Lines 111-113: “the key laboratory of new materials and facilities for rural renewable energy of the ministry of agriculture and rural affairs” involves some institution, the names should be capitalized. This comment should be considered throughout the text whenever the name of an institution is employed.

-        Lines 142-143: It is not clear what you mean by “In anaerobic digestion phase, the vacuum glass device was taken out every day after composite microorganisms pretreatment.”

-        Review the sentences on lines 162-164, 173-174, 177-178.

-        On lines 183-184, it is mentioned that the highest temperature is reached on day 6. However, Figure 1 shows that the temperature is maximum on day 5. You should clarify what day 0 means such that the results are well interpreted. Since at day 0 the temperature for both the control group and the composite microorganisms group is the same as the ambient temperature, day 0 represents the initial conditions of the experiments, which implies that the maximum temperature occurs after 5 days.

-        Lines 189-190 state that the temperature of the composite microorganisms was higher than the temperature of the control group in the first 9 days, but Figure 1 shows that for the entire experiment (14 days) this was always the case. Moreover, after the maximum on day 5, the temperature decreases at more or less the same rate until day 11.

-        The sentence on lines 192-193 “After the high-temperature period, its temperature began to decrease.” is really vague. Which is the high-temperature period? I believe that the entire subsection 3.1 needs attention.

-        On line 220 it is mentioned that the COD value decreased slowly after the peak, but Figure 3a shows a decrease from 9600 mg/L to 6000 mg/L in one day, so the decrease is not slow but rather sudden.

-        Lines 227-228: only the ethanol reaches the maximum on day 10, butyric acid reaches its maximum on day 12.

-        Lines 261-264 motivate that the gas production of the 4d group is lower than the one of the 3d and 9d groups because of the lower pretreatment temperature. How is this possible? Figure 1 shows that the temperature of the 4d group is only slightly smaller than the one of the 9d group, but it is higher than the one of the 3d group.

-        Lines 264-265 need revision

-        The colors of the gas production have been changed from the 3D plot to the 2D plot in Figure 5a

-        Figure 5c is blurry

-        The caption of Figure 5 needs revision (see “Rules of anaerobic digestion” and “gas production in abandon heat group”)

-        How were the parameters of the Gompertz model identified? The numerical values are reported but the methodology is not described.

-        Review the sentence on lines 365-366.

Comments on the Quality of English Language

Some of the sentences that need correction were already indicated in the main comments.  See also lines 218, 256, 344. There are also several typos at lines 79, 230, 272, 301-302, 331, 357.

Author Response

Dear reviewer of Fermentation,

  Thank you for your valuable comments on manuscript. We have revised the manuscript accordingly based on your review comments. Corresponding revises have been highlighted in red in the manuscript. The following amendments have been made to your comments:

  • The paper is about the treatment of corn straw, hence the conclusion in the abstract (lines 32-34) cannot be about “stabilizing corn straw” but about its treatme

Revision: Revises have been made in the manuscript according to your comments.

Line 32-34, the sentence has been revised to “In conclusion, this study highlights the potential of biological pretreatment with composite microorganisms followed by anaerobic digestion utilizing bio-heat as an effective method for treating corn straw.”

  • Lines 39-40: The sentence is not correct as the straw is converted to biogas through anaerobic digestion.

Revision: Revises have been made in the manuscript according to your comments.

Line 39-40, the sentence has been revised to “The straw can be effectively converted into biogas through anaerobic digestion.”

  • Lines 71-73: please define AMC and ANMC

Revision:

Line 72-75, “Zhu et al. [15], constructed the AMC (aerobic microbial consortium) aerobic complex strain, and the cumulative methane production of anaerobic fermentation was 233.09mL·g-1 VS and 242.56mL·g-1 VS after AMC and ANMC (anaerobic microbial consortium) biological pretreatment respectively, which increased by 6.89% and 11.23% compared with the control group.”

  • Lines 98-99: The name and affiliation of one of the authors of the present study is not the proper way of referencing. Unless there is a publication that can be cited, the sentence should be reformulated in a way similar to “In their previous work, the authors/ we …..”

Revision:Revises have been made in the manuscript according to your comments.

Line 101-103, the sentence has been revised to “In our previous works, the authors constructed a composite of microorganisms and found that the strain exhibited strong thermal effects during the pretreatment of crop straw, and the component structure of the straw changed correspondingly.”

  • Lines 111-113: “the key laboratory of new materials and facilities for rural renewable energy of the ministry of agriculture and rural affairs” involves some institution, the names should be capitalized. This comment should be considered throughout the text whenever the name of an institution is employed.

Revision: Revises have been made in the manuscript according to your comments.

Line 114-115, the sentence has been revised to “the Key Laboratory of New Materials and Facilities for Rural Renewable Energy of the Ministry of Agriculture and Rural Affairs.” 

The same errors in the manuscript have been revised.

Line 116-117, “China General Microbial Species Preservation and Management Center”

Line 119, “the College of Life Science, Henan Agricultural University”

Line 125-126, “the Key Laboratory of New Materials and Facilities for Rural Renewable Energy of Ministry of Agriculture and Rural Affairs”

  • Lines 142-143: It is not clear what you mean by “In anaerobic digestion phase, the vacuum glass device was taken out every day after composite microorganisms pretreatment.”

Revision: The experimental operations has been redescribed and corrections have been made at the appropriate places in the article.

Line 145-159, the sentence has been revised to “To examine the impact of varying pretreatment durations and the resultant heat generation on anaerobic digestion, the anaerobic digestion experiment was segmented into 14 runs, each corresponding to a different number of pretreatment days prior to anaerobic digestion. The pretreatment experiments were conducted using vacuum bottles. Following the pretreatment, inoculum was promptly introduced into the bottles, and deionized water was employed to regulate the TS concentration to 4%. Subsequently, the bottles were sealed for anaerobic digestion trials. Subsequently, the vacuum bottles were placed in a 35℃ incubator for a 30-day anaerobic digestion process. In this way it can be ensured that the material and bio-heat was brought to anaerobic digestion period. This 14 runs were set up as the pretreatment group. Then set up a control group with the same 14 runs which was carried into the anaerobic digestion stage after cooling in ambient temperature(18±2℃), this 14 groups were set up as the cooling group.The cooling group with 2,4,6,8,10 d of pretreatment was selected to compare the gas production effect with the pretreatment groups in the same pretreatment days. ”

  • Review the sentences on lines 162-164, 173-174, 177-178.

Revision:

Line 171-173, revised to “The 2mL of pretreatment solution was taken from the pretreatment unit every 2 days. Next the pretreatment solution was centrifuged at 5000 rpm for 10 min, then the supernatant were measured with a COD meter (KHCN-200A, Nanjing KEHUAN, China).”

Line 183-184, revised to “The cellulose, hemicellulose and lignin contents of corn straw were determined by the paradigm water washing method as employed by Zhang [25].”

Line 189-191, revised to “The column temperature was set at 60℃, the temperature of the thermal conductivity detector was set at 80℃. The carrier gas N2 was set at a flow rate of 25mL/min.”

  • On lines 183-184, it is mentioned that the highest temperature is reached on day 6. However, Figure 1 shows that the temperature is maximum on day 5. You should clarify what day 0 means such that the results are well interpreted. Since at day 0 the temperature for both the control group and the composite microorganisms group is the same as the ambient temperature, day 0 represents the initial conditions of the experiments, which implies that the maximum temperature occurs after 5 days.

Revision: The author’s intention was to reflect the fact that the initial temperature was ambient, with no external heat provided. Therefore the initial temperature of the experiment was the temperature on day 0. The “on day 6” expressed in the manuscript has been changed to “on day 5” such that the results are well interpreted.

Line 196-197, the sentence has been revised to “The highest temperature of composite microorganisms pretreatment group reached 56.2℃ on day 5.”

  • Lines 189-190 state that the temperature of the composite microorganisms was higher than the temperature of the control group in the first 9 days, but Figure 1 shows that for the entire experiment (14 days) this was always the case. Moreover, after the maximum on day 5, the temperature decreases at more or less the same rate until day 11.

Revision: The author’s description in this section was intended to convey that the temperature of composite microorganisms group began to decrease after the high-temperature period, but remained higher than that of control group. The control group continued to stabilize during the high-temperature phase, with a slight drop in temperature after day 5 followed by a small boost on day 10.

Line 203-206, the description of this section has been revised to “Throughout the 14 days of pretreatment process, the temperature of the composite microorganisms group was always higher than the control group, indicating that the metabolic activities of the microorganisms in composite microorganisms group were more rapid.”

  • The sentence on lines 192-193 “After the high-temperature period, its temperature began to decrease.” is really vague. Which is the high-temperature period? I believe that the entire subsection 3.1 needs attention.

Revision: The definition of the high-temperature period here is temperature above 40℃. Additional notes have been made in the manuscript.

Line 210-214, the description of this section has been revised to “The difference was that the control group experienced a relatively stable high-temperature period(above 40℃) after the warming period, with a slight drop in temperature after day 5 followed by a small boost on day 8-10. This also indicated that the degradation of corn straw without the intervention of composite microorganisms is slow and steady.”

  • On line 220 it is mentioned that the COD value decreased slowly after the peak, but Figure 3a shows a decrease from 9600 mg/L to 6000 mg/L in one day, so the decrease is not slow but rather sudden.

Revision: This section was misrepresented and has been revised in the manuscript.

Line 238-239, the sentence has been revised to “After the peak, the COD value was decreased and finally tended to be stable.”

  • Lines 227-228: only the ethanol reaches the maximum on day 10, butyric acid reaches its maximum on day 12.

Revision: This section was misrepresented and has been revised in the manuscript.

Line 245-246, the sentence has been revised to “Butyric acid and ethanol showed an increasing trend, ethanol reached the maximum value on day 10 and butyric acid reached on day 12.”

  • Lines 261-264 motivate that the gas production of the 4d group is lower than the one of the 3d and 9d groups because of the lower pretreatment temperature. How is this possible? Figure 1 shows that the temperature of the 4d group is only slightly smaller than the one of the 9d group, but it is higher than the one of the 3d group.

Revision: We have revised the description of this section.

Line 282-289, this section has been revised to “However, the gas production of the 4d group was lower than the other groups because of its lower pretreatment temperature compared to the 9d groups, resulting in less bio-heat accumulation, less active metabolism of the composite microorganisms, and less decomposition of soluble organic matter. As Compared to the 2d and 3d group, the reason for less gas production in the 4d group is that the acetic acid in the pretreatment substrate of the 2d and 3d group was much higher than other molecular acids and occupied the main metabolic substrate of methanogens.”

It remained to be stated that the maximum daily gas production of 4d group was 320mL lower than 3d group, and the cumulative gas production was 3765mL lower. The reason for this phenomenon is related to the metabolic evolution of the composite strains, another possibility is also related to the metabolic pathway of methanogenesis and further exploration of this phenomenon will be reflected in microbial diversity analyses in our study. This is another indication that the contribution of heat and degradation rate to methane production is not linearly superimposed.

  • Lines 264-265 need revision

Revision: Lines 286-289, revised to “As Compared to the 2d and 3d group, the reason for less gas production in the 4d group is that the acetic acid in the pretreatment substrate of the 2d and 3d group was much higher than other molecular acids and occupied the main metabolic substrate of methanogens.”

  • The colors of the gas production have been changed from the 3D plot to the 2D plot in Figure 5a

Revision: Line 331, Figure 5a has been modified in the manuscript.

  • Figure 5c is blurry

Revision: Line 348, Figure 5c has been modified in the manuscript.

  • The caption of Figure 5 needs revision (see “Rules of anaerobic digestion” and “gas production in abandon heat group”)

Revision: Line 381, the caption of Figure 5 has been revised to “Figure 5. Gas production pattern of corn straw anaerobic digestion after composite microorganisms pretreatment: (a) Plots of daily gas production; (b) Plots of methane content; (c) Plots of daily gas production in cooling group; (d) Plots of cumulative methane production.”

  • How were the parameters of the Gompertz model identified? The numerical values are reported but the methodology is not described.

Revision: The parameters of the Gompertz model in this manuscript were analyzed by the software Originpro 8.6 by nonlinear curve fitting. The supplementary note has been revised in section 2.3.

Line 168-169, “The parameters of the Gompertz model were obtained by the software Originpro 8.6 by nonlinear curve fitting.”

  • Review the sentence on lines 365-366.

Revision: Line 439-441, revised to “This conclusion has certain reference for the industrial application of composite microorganisms pretreatment method.”

This manuscript is a resubmission of an earlier submission. The following is a list of the peer review reports and author responses from that submission.

Round 1

Reviewer 1 Report

Comments and Suggestions for Authors

the english language is very poor and very difficult to understand. What is meant by Abandon heat? what by biothermal and biothermol?

Please make for the language corrections and submit again

Reviewer 2 Report

Comments and Suggestions for Authors

The manuscript is very difficult to read and understand. The experiments are poorly described and the materials and methods part lacks the necessary details to replicate the results. The scientific terms are wrongly used; therefore, the whole aim of the research is not clear.

Just few examples:

Anaerobic fermentation: The authors mean anaerobic digestion, since the end product is biogas. Methanogenesis is not fermentation but anaerobic respiration. The term anaerobic fermentation is used only in case the methanogenesis is inhibited and carboxylates are the intended products of the biotechnological application.

Compound bacteria: I think the authors mean a mixture of bacteria used for the pretreatment of straw; however, most of the used strains are not bacteria but fungi.

Gloeophyllum trabeum: It’s a fungus not a bacterium

Bacillus circulans: It’s a bacterium

Pseudomonas aeruginosa: It’s a bacterium

Phanerochaete chrysosporium: It’s a fungus

Trichoderma viride: It’s a fungus

Aspergillus niger: It’s a fungus

Coriolus versicolor: It’s a fungus

So it was an aerobic fungal treatment, considering the non-sterile straw used in this study, the whole process is similar to composting with the well-known heat production.

Just few more examples:

“The compound bacteria used in this paper were developed by the key laboratory of new materials and facilities for rural renewable energy of the ministry of agriculture and rural affairs. Including, Gloeophyllum trabeum, Bacillus circulans and Pseudomonas aeruginosa was purchased from China general microbial species preservation and management center, Phanerochaete chrysosporium, Trichoderma viride, Aspergillus niger and Coriolus versicolor were provided by the college of life science, Henan agricultural university. The compound bacteria was preserved after antagonistic test and expansion.”

You provide details about the institutes but not about the ratio of these (mainly fungal) strains and the cultivation conditions.

“The inoculum was obtained after anaerobic fermentation experiments of municipal sludge in key laboratory of new materials and facilities for rural renewable energy of ministry of agriculture and rural affairs.”

Again, the institute is quite irrelevant, but the term “municipal sludge” is very broad. Is it municipal sewage sludge? What kind of waste stream, what treatment, and what kind of experiment (no reference added) etc?

“In pretreatment process, the compound bacteria was inoculated into a liquid culture medium firstly.” (wrong grammar)

What kind of medium was used? Volume information is missing, as well as the amount of inoculum.

“Secondly, they were placed in a constant temperature shock incubator for  2 days of continuous activation (30 and 120r/min).

What is a “temperature shock incubator”?

“In anaerobic fermentation process, the vacuum glass device was taken out every day after compound bacteria pretreatment.” Every day? (Do you mean glass vacuum filtration device?)

“Then the vents of vacuum glass device was sealed to form an anaerobic condition and placed in 35 constant temperature incubator for continuous culture for 30 days.”

Why it was not flushed with N2? Was it shaken?

“The group which was carried into the anaerobic fermentation stage after cooled was set up as abandon heat 135 (AH) group.”

I can’t figure out the meaning of this sentence.

How was the gas production measured? Was it normalized? Why do you need the TS/VS of the “the digestion liquid”. Hopefully, it was not used for calculating the gas yield and methane yield in Fig 5.

The figure use the term “production” but the unit (mL per gram VS) suggests yield or specific gas production, which is probably not correct in Fig5a and c (since it was a batch process, you cannot calculate daily yield, there was no daily feeding of a substrate)

At the currents stage, it is very difficult to read or interpret the manuscript.